# Nuclear Reorganization in Hippocampal Granule Cell Neurons from a Mouse Model of Down Syndrome: Changes in Chromatin Configuration, Nucleoli and Cajal Bodies

**DOI:** 10.3390/ijms22031259

**Published:** 2021-01-27

**Authors:** Alba Puente-Bedia, María T. Berciano, Olga Tapia, Carmen Martínez-Cué, Miguel Lafarga, Noemí Rueda

**Affiliations:** 1Department of Physiology and Pharmacology, Faculty of Medicine, University of Cantabria, 39011 Santander, Spain; alba.puente@unican.es (A.P.-B.); martinec@unican.es (C.M.-C.); 2Department of Molecular Biology, “Red sobre Enfermedades Neurodegenerativas (CIBERNED)” and University of Cantabria-IDIVAL, 39011 Santander, Spain; maria.berciano@unican.es; 3Instituto de Investigación Sanitaria Valdecilla (IDIVAL), “Red sobre Enfermedades Neurodegenerativas (CIBERNED)” and Universidad Europea del Atlántico, 39011 Santander, Spain; otapia@idival.org; 4Department of Anatomy and Cell Biology, “Red sobre Enfermedades Neurodegenerativas (CIBERNED)” and University of Cantabria-IDIVAL, 39011 Santander, Spain

**Keywords:** Ts65Dn, hippocampus, Down syndrome, nucleolus, Cajal body, chromatin, NORs

## Abstract

Down syndrome (DS) or trisomy of chromosome 21 (Hsa21) is characterized by impaired hippocampal-dependent learning and memory. These alterations are due to defective neurogenesis and to neuromorphological and functional anomalies of numerous neuronal populations, including hippocampal granular cells (GCs). It has been proposed that the additional gene dose in trisomic cells induces modifications in nuclear compartments and on the chromatin landscape, which could contribute to some DS phenotypes. The Ts65Dn (TS) mouse model of DS carries a triplication of 92 genes orthologous to those found in Hsa21, and shares many phenotypes with DS individuals, including cognitive and neuromorphological alterations. Considering its essential role in hippocampal memory formation, we investigated whether the triplication of this set of Hsa21 orthologous genes in TS mice modifies the nuclear architecture of their GCs. Our results show that the TS mouse presents alterations in the nuclear architecture of its GCs, affecting nuclear compartments involved in transcription and pre-rRNA and pre-mRNA processing. In particular, the GCs of the TS mouse show alterations in the nucleolar fusion pattern and the molecular assembly of Cajal bodies (CBs). Furthermore, hippocampal GCs of TS mice present an epigenetic dysregulation of chromatin that results in an increased heterochromatinization and reduced global transcriptional activity. These nuclear alterations could play an important role in the neuromorphological and/or functional alterations of the hippocampal GCs implicated in the cognitive dysfunction characteristic of TS mice.

## 1. Introduction

Down syndrome (DS), the most frequent genetic cause of cognitive dysfunction [1] is caused by a partial or complete triplication of the human chromosome 21 (Hsa21). To study the mechanisms implicated in the neurobiological and cognitive alterations found in DS and to develop therapeutic approaches to reduce or prevent these impairments, several mouse models of DS, trisomic for different sets of Hsa21 orthologous genes, have been generated [2,3,4,5]. Among them, the Ts65Dn mouse (TS) harbors a translocation of a segment of Mmu16 that contains 92 genes orthologous to those found in the Hsa21 [2,6]. It also exhibits numerous neuroanatomical, neurochemical, and behavioral DS phenotypes similar to those found in individuals with DS [3,4,5].

In DS and in the TS mouse, hippocampal-dependent learning and memory is particularly affected due to anatomical and functional alterations of this structure [3,7]. Reduced hippocampal volume is associated with a smaller density of hippocampal granular cells (GCs) [8,9,10]. This hypocellularity is mainly caused by alterations in pre- and post-natal neurogenesis [7,8,10]. In addition to hypocellularity, abnormal dendritic tree arborization, spine density and morphology, and synaptic density and connectivity have also been found in the granular cell layer (GCL), further altering proper hippocampal function [11,12,13,14].

The cell nucleus is organized into structural and functional compartments involved in transcription, DNA replication and repair, and RNA processing. They include chromosome territories, the distinct nuclear sub-volumes occupied by each interphase chromosome, and the interchromatin region, which contains several nuclear structures with roles in gene expression and RNA processing, such as the nucleolus, Cajal bodies, and nuclear speckles of splicing factors [15,16,17,18].

Chromosome territories are nonrandomly arranged within the interphase nucleus, and a precise 3D organization of the genome is required for establishing cell-specific gene expression programs [16,19]. Underscoring the closely intertwined relationship between genome topology and gene regulation, alterations of the nuclear architecture that perturb intrachromosomal and interchromosomal interactions can disturb gene expression and lead to diseases [19]. Chromosome territories include domains of transcriptionally silenced heterochromatin, which harbor specific proteins and histone repressive marks (H3K9me, H4K27me3; H4K20me3), as well as transcriptionally active euchromatin enriched in acetylated nucleosomal histone marks and components of the transcription machinery.

The nucleolus is an archetype of nuclear compartment that is assembled around rDNA sequences called nucleolar organizer regions (NORs), where repeated rRNA genes are arranged in tandem [20,21,22]. Each ribosomal gene transcription unit encodes for three categories of mature rRNAs, 28S, 18S and 5.8S, produced from a single 47S rRNA precursor which is transcribed by RNA polymerase I activity (RNA pol I). The nucleolus is the site of ribosome biogenesis, where pre-rRNA synthesis, pre-rRNA processing, and the assembly of pre-ribosomal particles takes place [15,21,23,24].

Because of the essential role of the nucleolus in sustaining the protein synthesis machinery in developing and mature neurons, alterations in its structure and/or function can disrupt neurodevelopment, and may also be involved in the pathophysiology of several neurodegenerative disorders, including cerebellar ataxias, Alzheimer Disease (AD), Parkinson’s disease, and spinal muscular atrophy (SMA) [25,26,27,28,29].

Cajal bodies (CBs) are dynamic nuclear structures of the interchromatinic region, discovered in mammalian neurons by Cajal [18,30,31,32,33]. They concentrate coilin, a structural protein and molecular marker of CBs, the survival of motor neuron (SMN) protein, spliceosomal small nuclear ribonucleoproteins (snRNPs), small nucleolar RNAs (snoRNAs) and CB-specific snRNAs (scaRNAs). Moreover, like the nucleolus, CBs contain fibrillarin, a protein involved in pre-rRNA processing [18,30,32]. Current conceptions of CBs assembly support the view that they form on active histone and U snRNAs or snoRNA gene loci, and that this CB-dependent gene positioning contributes to genome organization [33,34]. CBs are transcription-dependent nuclear compartments essentially involved in the biogenesis of spliceosomal snRNPs which are required for pre-mRNA processing in the spliceosome [32,35,36]. In mammalian neurons, the CB number positively correlates with neuronal mass and transcriptional activity [37,38]. Depletion of CBs, in neurodegenerative diseases such as SMA, perturbs pre-mRNA splicing, and contributes to the pathophysiology of the disease [29,39].

Recent advances in genomics and epigenomics reveal that the 3D spatial organization of the genome contributes to the regulation of gene expression through complex long-range intrachromosomal and interchromosomal interactions, which are often mediated by the formation of chromatin loops. In particular, newborn maturing postmitotic mammalian neurons that exit from cell cycle must acquire, in the early G1/G0 phase, the stable 3D genome organization characteristic of each neuronal type. This includes chromosome movements to achieve the spatial rearrangement of chromosome territories, the spatial tethering of NORs sequences of rDNA resulting in nucleolar fusion, and the acquisition of the epigenetic pattern of chromatin configuration with the specific heterochromatin (transcriptionally repressed) and euchromatin (transcriptionally active or poised for activation) pattern domains [21,40,41].

In this context, it has been demonstrated that the extra Hsa21 causes a displacement and higher compaction of other chromosome territories in trisomic cells [42]. In addition, the supernumerary Hsa21 or Mmu16 fragment also creates gene expression dysregulation domains and modifies the histone mark profiles in DS and TS cells [43]. Significantly, in the TS mouse, analysis of the gene expression in different organs including the brain has revealed that while some of the triplicated genes are overexpressed others are downregulated or kept below normal levels of expression [44,45], suggesting the presence of additional tissue-specific regulatory mechanisms [44]. Finally, in addition to the altered expression of trisomic genes, the excessive gene dosage in DS also dysregulates the global transcription of other disomic genes [43,46,47,48,49]. Thus, the genomic dose imbalance causes a complex regulation of gene expression and epigenetic alterations, such as DNA and histone modifications, which have been implicated in several DS phenotypes such as memory impairment, premature aging, and neurodevelopmental defects [5,50].

Because of its essential role in hippocampal learning and memory, alterations in the 3D reorganization of the genome and/or in the nuclear structure and function of hippocampal GCs could contribute to the neuromorphological or functional alterations observed in this neuronal population. In particular, these alterations may be implicated in the cognitive dysfunction that the TS mouse presents. The aim of this study is to analyze the effect that the 3D reorganization of the genome induced by the partial trisomy of Mmu16 in the GCs of the TS mouse has on the nuclear size and organization of the nuclear compartments involved in transcription and mRNA and rRNA processing. We will primarily focus on (I) the epigenetic changes in chromatin structure and its potential influence on global transcriptional activity, (II) the reorganization of NORs, and its effect on the fusion pattern of nucleoli, and (III) the behavior of CBs.

## 2. Results

### 2.1. Reduced Nuclear Size and Increased Heterochromatinization Are Nuclear Features in TS GCs

The impact of alterations in the 3D-organization of chromosomes and their associated changes in the epigenetic programs in GCs was studied in TS mice. First, we analyzed variations in the nuclear size and the heterochromatin compartment. Dissociated neuronal preparations of the dentate gyrus GCL from wild-type (hereafter referred to as control) mice co-stained with DAPI and propidium iodide, two cytochemical markers for DNA and RNA, respectively, revealed the well-preserved morphology of whole GC perikarya. Thus, the cell nucleus showed several prominent masses of condensed heterochromatin and appeared to be surrounded by a narrow RNA-rich cytoplasmic band stained with propidium iodide (Figure 1A).

Next, we performed immunolabeling for the repressive histone modification H4K20me3, a heterochromatin marker, which is involved in gene silencing and also contributes to the cellular senescence program [51,52,53]. As is illustrated in Figure 1B,C, the GCs of TS mice exhibited a clear increase of the heterochromatin regions immunolabeled for the histone H4K20me3 relative to control neurons. Morphometric measurements of the nuclear surface area of control and TS GCs on histone H4K20me3 immunostained preparations revealed a significant reduction of nuclear size, a parameter related to the heterochromatinization level and transcriptional activity [54,55], in TS mice when compared with control animals (Figure 1D). Consistent with the reduction of the nuclear size, morphometric estimation with Image-J software (NIH, USA) of the total nuclear area occupied by H4K20me3-positive chromatin regions confirmed a significant increase of heterochromatin in the GCs from TS mice as compared with control mice (Figure 1D). The upregulation of the amount of histone H4K20me3 in hippocampal tissue lysates from TS mice relative to control animals was validated by Western blotting (Figure 1E).

Moreover, the expression levels of two genes, *Mecp2* and *H1f0*, encoding methyl-CpG-binding protein 2 (MeCP2), which has a great affinity for methylated DNA, and the H1.0 linker histone, respectively, were significantly increased in the hippocampal RNA extracts from TS mice relative to control animals (Figure 1F). Interestingly, both abundant nuclear proteins are important for the maintenance of higher-order chromatin structure, particularly of heterochromatin domains [56,57]. In this line, electron microscopy analysis of the GCL confirmed the higher abundance of heterochromatin clumps in the tightly packed GC nuclei from TS mice (Figure 1G,H).

### 2.2. Chromatin Reorganization in TS GCs Induces a Decrease in Global Transcription Rate

After establishing that the reorganization of the genome architecture in TS GCs is accompanied by increased heterochromatinization, we investigated the impact of these chromatin changes on global transcriptional activity. We analyzed the lysine acetylation pattern in the nucleosomal histone H4 as a marker of “open” transcriptionally active chromatin [58]. The immunostaining for the acetylated histone H4 showed a higher intensity of fluorescent signal in control GC nuclei than in TS ones (Figure 2A,B). Densitometric analysis of signal intensities with ImageJ software (NIH, USA) confirmed the reduction of this acetylated histone H4 in TS GC nuclei relative to control nuclei (Figure 2C). Moreover, the significant reduction of acetylated histone H4 levels in TS mice was validated by Western blotting of hippocampal lysates (Figure 2D).

To determine the global transcriptional activity, we performed an in situ transcription assay based on the incorporation of 5′-FU into nascent RNA 45 min after the intraperitoneal injection of this halogenated nucleotide precursor. The incorporation pattern of 5′-FU showed the typical nuclear punctate pattern, corresponding to transcription factories [59,60], in addition to a diffuse nucleoplasmic signal (Figure 2E,F). Densitometric analysis of 5′-FU fluorescent signal in confocal images of the total nuclear surface area revealed a significant increase in control GC nuclei as compared with TS GC nuclei (Figure 2G), supporting a reduced transcription rate in TS GCs.

Next, we investigated whether the reduction of transcription was accompanied by a reorganization of the translation machinery in the GC cytoplasm from TS mice. As an initial cytochemical approach, we observed an apparent decrease in the cytoplasmic RNA signal with the propidium iodide staining for nucleic acids, suggesting a reduced cytoplasmic availability of mRNAs for translation (Appendix A).

### 2.3. The Reorganization of Chromosomes in TS GCs Is Associated with Reduced Nucleolar Fusion, Resulting in a Higher Number of Nucleoli

Since the DS trisomy involves positional changes in the 3D-organization of the genome, presumably affecting the NOR-containing chromosomes, we investigated the impact of this genome rearrangement on the behavior of nucleoli in TS GCs. It is well known that post-mitotic mammalian neurons start with several small nucleoli which fuse together and generally form only one or two larger ones [37,40]. In the case of post-mitotic mouse GCs, they are diploid with six NOR-bearing chromosomes [21]. When GC precursors leave the last telophase and enter a quiescent G0 state, they may initially form multiple nucleoli, one for each NOR, before starting nucleolar fusion events during GC differentiation.

The mean number of nucleoli per nucleus in mature GCs was estimated in dissociated GCs immunostained for nucleolin, an abundant nucleolar protein that interacts with rDNA and pre-rRNA and is involved in the spatial organization of rDNA arrays [24,26,61]. Nucleolin immunostaining revealed small and round-shaped nucleoli in both control and TS GCs, which are commonly associated with large masses of perinucleolar heterochromatin, intensely counterstained with DAPI (Figure 3A,B). Importantly, the number of nucleoli was significantly higher in TS GCs than in control ones (Figure 3A,B,E). According with the essential role of nucleolin in nucleolar assembly [61,62], this increase in the average number of nucleoli per GC was accompanied by a moderate increase in the expression level of *Ncl* gene in RNA extracts of the hippocampi from TS mice (Figure 3E). However, non-significant changes in the expression of nucleolin protein were observed in TS mice hippocampi as compared with control samples (Appendix A). Ag-NOR protein staining for rDNA sites [22] confirmed the increase in the number of nucleoli in GCs from TS mice (Figure 3C,D,F). Moreover, this Ag-staining procedure distinguished two types of nuclear bodies: typical sharply defined nucleoli, where active NORs are concentrated [22], and between one and three smaller foci, which corresponded to extranucleolar rDNA sites (Figure 3C,D). The number of the Ag-stained foci increased significantly in TS GCs as compared with control neurons (Figure 3F). The augmentation in the number of both nucleoli and Ag-stained foci supports the hypothesis that the nucleolar fusion pattern is altered in TS GCs.

Ultrastructural analysis of nucleoli from both control and TS GCs showed a small size, rounded morphology, and a compact configuration with a predominant granular component (Figure 3G,H). Nucleolus associated domains (NAD) enriched in repressive heterochromatin were prominent in both TS and control GCs (Figure 3G,H).

To determine whether an increase in the number of nucleoli was correlated with variations in transcriptional activity of rDNA and several genes encoding major nucleolar proteins, such as fibrillarin (*Fbn1)*, nucleophosmin/B23 (*Npm1*) and UBF (*Ubft*), we performed RT-qPCR on hippocampal RNA extracts from control and TS mice. Whereas 45S pre-rRNA is very short-lived and is considered to be an indicator of RNA pol I-dependent transcriptional activity of ribosomal genes [63], 18S and 28S rRNAs correspond to mature forms. Non-significant changes in the expression levels of 45S pre-rRNA, 18S 5′ junction and 28S 5′ junction intermediate precursors, and mature 18S and 28S rRNAs were detected in TS mice as compared to control ones (Figure 4A,B). This suggests that the transcription rate of 45S rRNA and its nucleolar processing were preserved in TS GCs. The next step was to investigate the expression levels of genes encoding UBF, fibrillarin and nucleophosmin/B23, three important proteins for pre-rRNA synthesis and processing as well as for the assembly of pre-ribosomal particles [15,24]. According with the preserved 45S pre-rRNA transcription in TS hippocampus, no significant differences in *Ubft* mRNA expression, encoding a key transcription factor of ribosomal genes, were found between control and TS hippocampal RNA extracts (Figure 4C). Similarly, no significant changes were observed in the expression of *Fbn1* and *Npm1* genes between control and TS hippocampi (Figure 4C).

### 2.4. The Nuclear Reorganization in GCs of the TS Mouse Results in Reduced CB Number and Coilin Redistribution

Given that CBs are transcription-dependent nuclear compartments involved in essential nuclear functions, particularly in pre-mRNA and pre-rRNA processing [18,31,32,33,35], we investigated the behavior of these nuclear bodies in GCs of the TS mouse. First, in dissociated GCs from control and TS mice, we estimated the CB number in preparations immunostained for coilin, a CB marker, counterstained with propidium iodide (Figure 5A–F). A significant decrease in the mean number of CBs per GC was detected in TS mice when compared with control animals (Figure 5G). Moreover, a fraction of GCs from the TS mice, approximately 12%, showed perinucleolar caps or “rossetes” of coilin [29] instead of typical spherical CBs, whereas this relocalization of coilin was only found in about 2% of control GCs (Figure 5G). No significant differences in the expression level of coilin mRNA were detected by RT-qPCR between control and TS mice in RNA extracts from the hippocampi (Figure 5G), suggesting a redistribution of coilin protein in trisomic mice rather than a downregulation of *Coil* gene expression.

Next, to determine the presence of spliceosomal snRNPs in CBs we performed double immunolabeling for TMG-cap, which recognizes the 5′ cap of spliceosomal snRNAs, and coilin. Whereas both molecular constituents colocalized in typical CBs from both control and TS mice, indicating that they are canonical CBs, the fluorescent signal of TMG-cap was very weak or absent in perinucleolar rossetes (Figure 5H–K insets). Finally, the localization of coilin in CBs and perinucleolar rosettes was confirmed by immunogold electron microscopy which illustrated the distribution of coilin in the coiled threads of typical CBs from control mice (Figure 5J), and in perinucleolar caps from TS mice (Figure 5K).

## 3. Discussion

Our results indicate that the partial trisomy of a segment of Mmu16 in TS mice induces important changes in the nuclear architecture of GCs, particularly in the nuclear compartments involved in transcription and pre-rRNA and pre-mRNA processing. This nuclear reorganization of GCs influences: (I) the epigenetic regulation of chromatin structure and function, with increased heterochromatin and a global reduction of transcription, (II) the nucleolar fusion pattern, with increased number of nucleoli, and (III) the molecular assembly of the CBs involved in pre-mRNA splicing.

Notwithstanding the influence of the dosage imbalance of gene expression in this mouse model of DS [64], we propose that the triplication of a large segment of Mmu16 into the genome of GCs affects *per se* their 3D genome organization. This could result in physical alterations in both gene positioning and the intrachromosomal and interchromosomal interactions mediated by chromatin loops. Given the closely intertwined relationship between genome topology and gene regulation, changes in the gene positioning and chromatin interactions in DS trisomy could lead to altered gene expression, resulting in a global downregulation of transcription observed here in the TS GCs. This effect may contribute to the neuromorphological and functional alterations found in the hippocampal GCs [8,12,13,14]. The alterations in the nuclear architecture of TS GCs should occur during maturation, when postmitotic neurons undergo a spatial reorganization of chromosomes in order to achieve the specific 3D nuclear architecture of differentiated mature neurons [41]. Thus, the establishment and maintenance of neuron-specific gene expression programs in hippocampal GCs allow for their specialized functions, such as the processing of information and the formation of memories.

The hippocampal GCs of the dentate gyrus have a small nucleus with prominent heterochromatin masses of transcriptionally repressed heterochromatin and compact small nucleoli [65]. The reduction of nuclear size in TS GCs is consistent with their increased heterochromatin domains and reduced global transcription. Nuclear size has been positively correlated with DNA content and global transcription rate [37,54,66]. Interestingly, a reduction of both nuclear size and RNA synthesis has been reported in differentiating neuronal progenitors in the absence of the methyl-CpG-binding protein 2 (MeCP2) [66]. This protein recognizes unmethylated and methylated regions of the genome and, acting as an epigenetic regulator of gene expression, can both repress and activate specific genes depending on the context [67]. In agreement with the close relationship between epigenetic regulation of chromatin and nuclear size in TS GCs, we found that increased levels of histone H4K20me3, a marker of repressive heterochromatin domains [52,53], and decreased expression of acetylated histone H4 both correlated with reduced nuclear size. Moreover, two genes, *Mecp*2 and *H1f0*, involved in the organization of heterochromatin are upregulated in the hippocampi of TS mice [56,57,67]. In this context, chromatin compaction in heterochromatin should reduce the nuclear subvolume occupied by chromosome territories, leading to decreased nuclear size, as has been reported in some neurodegenerative disorders such as SMA [55].

The global reduction of the transcription rate in TS GCs detected with the in situ 5′-FU transcription assay is noteworthy. The decreased incorporation of 5′-FU into nascent RNA clearly occurs in active euchromatin domains, supporting the fact that downregulated genes mainly correspond to protein coding genes. However, the contribution of chromatin non-coding genes cannot be ruled out. Consistent with the run-on transcription assay, we have detected a reduction of the acetylated histone H4, a marker of “open” transcriptionally active euchromatin [58], by immunofluorescence and Western blotting.

Despite the high complexity of gene expression in aneuploid mammalian cells, a common feature is decreased proliferative capacity [64]. In particular, mouse embryonic fibroblasts trisomic for Mmu16 show an upregulation of the genes involved in the cellular stress response, and downregulation of cell cycle transcripts, including mitosis, DNA replication, and chromosome segregation [64]. This transcriptional response can impact the neurogenesis of hippocampal GCs resulting in the reduction of mature neurons previously reported in the TS mouse model of DS [8,68].

Regarding the particular impact of the Mmu16 trisomy on global gene expression in the central nervous system, Kahlem et al. [44] reported increased transcripts in the majority of trisomic genes in the cerebral cortex and cerebellum of TS mice, supporting an upregulation which is proportional to the gene copy number. However, some trisomic genes are downregulated, suggesting the participation of additional regulatory mechanisms [44]. Beyond the expression levels of trisomic genes in the TS mouse, which represents a small fraction of the mouse genome, our results support a global reduction of transcriptional activity. We propose that two factors are essential in this transcriptional response. First, the changes in nuclear architecture and gene positioning induced by Mmu16 trisomy may disturb normal gene interactions, resulting in a downregulation of gene expression [19]. For example, downregulation of several disomic genes implicated in the survival and maintenance of neuronal pathways has been found in CA1 pyramidal neurons of TS mice. Second, a common feature of aneuploid mammalian cells is the altered regulation of cellular stress response genes [64]. In this regard, it is well known that cellular stress commonly induces a genome-wide transcriptional repression of thousands of RNA pol II activity-dependent genes, excluding stress-related genes that are upregulated [69].

Additionally, the altered expression of several orthologous Hsa21 genes, triplicated in the TS mouse, may contribute to increased heterochromatinization, or the decrease in global transcription found in the GCs of this mouse model. Among them, *Hlcs* catalyzes the biotinylation of some histones and induces chromatin condensation and the repression of genes [70,71]. Other triplicated genes such as *Dyrk1A* and *Runx1* codify proteins that may alter the epigenetic architecture of the nucleus [72,73].

In relation to the nucleolar behavior of GCs from the TS mouse, the increase in the number of nucleoli is remarkable. Two important factors may influence the determination of the number of nucleoli in maturing GCs: the number of NORs, and the nucleolar fusion pattern. In eukaryotic cells, beginning at telophase, the transcription of rRNA genes resumes and nucleoli start to reform around individual NORs, initially forming multiple nucleoli. They fuse together, resulting in the assembly of one or a few larger mature nucleoli, each containing several NORs [21,24,40]. In euploid mice there are six NOR-bearing acrocentric chromosomes (12, 15, 16, 17, 18, 19), and their NORs are positioned close to telomeres [21]. Interestingly, while the triplication of Hsa21 provides an additional NOR in DS, the trisomic Mmu16 segment of the TS mouse does not include an additional NOR [74]. Therefore, the increase in TS GC nucleoli is independent of the presence of a supernumerary NOR.

Regarding nucleolar fusion, its cellular and molecular mechanisms are not well known, but there is evidence that this cellular process must involve an important spatial reorganization of NOR-bearing chromosomes within the nucleus [21]. Indeed, cell-life experiments in human cells during the early G1 phase have shown ATP-dependent chromosome displacements of 3–5µm, and also that nucleoli usually independently move away from each other [75,76]. However, pairs of nucleoli coordinate and slow down their movements when they are to be fused. Taking into consideration that nucleolar rDNA arrays are physically tethered to chromatin fibers of NOR-bearing chromosomes, nucleolar motion must lead to local spatial reorganization of chromatin [76]. Moreover, it has been proposed that the nucleoplasm viscosity around the nucleolus, as well as the interaction of the nucleolus with the surrounding chromatin, sets the time scale for nucleolar fusion kinetics [76].

In this context, we propose that partial trisomy-associated alterations in nuclear architecture and the epigenetic state of chromatin in differentiating TS GCs interferes with the normal pattern of nucleolar fusion, resulting in an increase in the number of nucleoli. In addition to rRNA transcription, nucleolin plays a key role in nucleolar assembly [61,62,77]. We suggest that the preserved expression of nucleolin mRNA and protein may contribute to the maintenance of an elevated number of nucleoli in mature TS GCs. Moreover, although nucleolin is a multifunctional protein with a predominant nucleolar localization, there is a nucleoplasmic pool [26,78] which could potentially be involved in the biogenesis of the supernumerary nucleoli.

Importantly, whereas most large size mammalian neurons are mononucleolated, small size neurons with lower transcriptional activity tend to be multinucleolated. This suggests a negative correlation between transcriptional activity and the number of nucleoli in mammalian neurons [37,38]. It is thought that the presence of a single nucleolus allows the concentration of active ribosomal genes and the molecular machinery for rRNA transcription and processing in a unique nucleolar domain, which potentially enhances the molecular interactions required for nucleolar activity.

Supporting the hypothesis of the spatial reorganization of NOR-bearing chromosomes in trisomic GCs, we have found changes in the normal distribution of Ag-NORs stained with a silver nitrate procedure that recognizes some argyrophilic proteins associated with rDNA arrays [21,22]. In addition to the presence of typical Ag-NORs associated with nucleoli [21], we observed a significant increase of smaller extranucleolar foci of Ag-NORs, suggesting that partial Mmu16 trisomy induces positional changes in some NOR-bearing chromosomes.

Our results show non-significant changes in the expression of both 45S pre-rRNA and mature 18S and 28S rRNAs in hippocampal TS mice compared with control ones, suggesting that ribosomal gene transcription is preserved in TS GCs. Consistent with these findings, no variations in the expression of genes encoding three key nucleolar proteins, UBF, fibrillarin and nucleophosmin, were found between control and TS hippocampal RNA extracts. The maintenance of ribosomal gene transcription in TS GCs may reflect a reactive response of the nucleolus to the global reduction of the transcription of protein-coding genes, in an attempt to preserve ribosome biogenesis and translational activity. A similar reactive compensatory nucleolar response has been reported in (i) SMA spinal motor neurons, (ii) sensory ganglion neurons under experimental conditions of proteotoxic stress and (iii) certain neuronal populations from AD patients [27,29,79]. It is also noteworthy that the increase in the number of nucleoli in TS GCs does not correlate with a parallel increase of nucleolar transcription.

Another important issue is the changes in the organization of CBs in TS GCs, with two crucial manifestations: a reduction in the number of canonical CBs, and an increased incidence of perinucleolar caps of coilin which are free of snRNPs. CBs are very dynamic transcription-dependent nuclear structures [18,32,33,35]. In post-mitotic neurons with high transcription and splicing demands, CBs play an essential role in the biogenesis of spliceosomal snRNPs [30,32,80]. Several experimental or pathological conditions including cellular stress, inhibition of transcription, DNA damage, and splicing dysfunction induced by SMN deficiency in SMA motor neurons induce disruption and loss of CBs [25,29,81,82]. In this context, the depletion of CBs is consistent with both cellular stress and a drop in the global transcription rate and, consequently, in the neuronal demand for pre-mRNA splicing in TS GCs. Moreover, current conceptions of CB assembly indicate that they are formed on active U snRNA or snoRNA and histone gene loci [33,34], suggesting that the reduced transcription of these genes can directly contribute to the decrease in the number of CBs.

The increased formation of perinucleolar caps of coilin suggests a defective assembly of CBs which is dependent on post-translational modifications of CB proteins such as SMN and coilin [80,83]. Coilin methylation is a key regulatory factor for CB assembly [84], and hypomethylated coilin has been reported to relocalize in perinucleolar caps upon inhibition of protein methylation or SMN depletion in mammalian neurons [29,85].

## 4. Conclusions

In conclusion, in this study we demonstrate that the three copies of a fragment of Mmu16 in the TS mouse alters the nuclear architecture (increase of both heterochromatin domains and the number of nucleoli, and loss of CBs) of their hippocampal GCs. The organization of CBs and nucleoli is emerging as a reliable nuclear indicator of neuronal homeostasis, and of its dysfunction under neurodegenerative conditions such as AD [27,28,86]. Because of the fundamental role of CBs in pre-mRNA processing, the alterations of these nuclear structures in GCs of TS mice may disturb mature mRNA synthesis and translation, affecting the maintenance of neuronal homeostasis, particularly the function of dendrites and synapses. Furthermore, the 3D genome reorganization of the hippocampal GCs of TS mice produces an epigenetic dysregulation of chromatin that results in increased heterochromatinization and reduced global transcriptional activity. Besides its influence on trisomic genes, this response may alter the expression of other disomic genes with essential functions in neuronal proliferation, maintenance, or survival. Because GCs have a fundamental role in hippocampal-dependent learning and memory [87], alterations in their nuclear compartments or 3D genome organization may play an important role in the altered neurogenesis and contribute to the cognitive deficits of this mouse model of DS.

## 5. Materials and methods

### 5.1. Animals

This study was approved by the Cantabria University Institutional Laboratory Animal Care and Use Committee and was performed in accordance with the Declaration of Helsinki and the European Communities Council Directive (86/609/EEC of 24 November 1986). TS and control mice were generated and karyotyped using real-time quantitative PCR (qPCR) as previously described [68]. In all the experiments, 6-month-old TS mice were compared with their euploid control littermates.

### 5.2. Immunofluorescence and Confocal Microscopy

For immunofluorescence, at least three animals of each karyotype were anesthetized and perfused under deep anesthesia with paraformaldehyde 3.7% in PBS, and post-fixed for 1 h. Their brains were removed and coronally sliced in a vibratome (200 μm thick). Small fragments from the GCL of the hippocampus were dissected out and transferred to a drop of PBS on a siliconized slide. Squash preparations of dissociated GCs were performed following the protocol described by [37]. The samples were then sequentially treated with 0.1 glycine in PBS for 20 min, 0.5% of Triton X-100 in PBS for 45 min, and 0.05% of Tween20 in PBS for 5 min. The samples were then incubated overnight at 4 °C with the primary antibody, washed with 0.05% of Tween20 in PBS, incubated 45 min in the specific secondary antibody conjugated with FITC or TexasRed (Jackson, Laboratories, USA Jackson, West Grove, PA, USA), and mounted with Vectashield with DAPI (Vector Laboratories, Inc., Burlingame, CA, USA). Some samples were counterstained with propidium iodide.

Confocal images were obtained with a LSM510 (Zeiss, Germany) laser scanning microscope using a 63x Oil (1.4 NA) objective. In order to avoid overlapping signals, images were obtained by sequential excitation at 488 nm and 543 nm to detect FITC or TexasRed, respectively. The images were processed using Adobe Photoshop CS4 software (Adobe Systems Inc, California, CA, USA).

The following primary antibodies were used: polyclonal rabbit anti-acetyl histone H4 (06-598, Upstate), polyclonal rabbit anti-tri-methylated histone H4 at K20 (07-463, Upstate), anti-coilin 204/10 (204.3 serum), anti-TMG-cap (NA02A, Oncogene) and anti-nucleolin (ab22758, Abcam). Morphometric and quantitative analyses were performed using ImageJ software (US National Institutes of Health, http://imagej.nih.gov/ij). At least 100 GCs per animal were sampled in all the analyses. The nuclear area of the GC was determined using DAPI staining. Nuclear perimeters were traced by hand and the areas were measured with Image J default tools to calculate the shape descriptors. Acetylated histone-H4 fluorescence intensity was quantified within the nuclear perimeter of the GC. Quantification of the area occupied by tri-methylated histone-H4 was performed in single-plane images. First, the nuclear area was determined and then the regions occupied by tri-methylated histone-H4 were delimited by applying an automatic local threshold. For the quantitative analysis of the number of CBs and nucleoli, coilin and nucleolin positive spots were counted, respectively, on serial confocal sections of the whole nucleus.

### 5.3. In Situ Transcription Assays with 5′-Fluorouridine

Active transcription sites were labelled by incorporation of 5′-fluorouridine (5′-FU) into nascent RNA as previously described [60]. The mice were given an intraperitoneal injection of 5-FU (Sigma, Darmstadt, Germany) at a dose of 10 μL/g of a stock solution of 0.4 M 5-FU in 0.9% saline. The mice were euthanized 45 min post-injection and fixed by perfusion with 3.7% paraformaldehyde in HPEM buffer (2× HPEM: Hepes, 60 mM; Pipes, 130 mM; EGTA, 20 mM; and MgCl_2_·6H_2_O, 4 mM) containing 0.5% Triton X for 10 min. Their brains were removed, post-fixed with a perfusion buffer for 1 h and washed in HPEM. Then, mechanical GCs dissociation was performed as described above. The incorporation of 5′-FU into nascent RNA was detected with mouse monoclonal anti-BrdU (Clone BU-33, Sigma, Darmstadt, Germany) for 1 h at 37 °C. Then, the samples were washed with 0.01% Tween 20 in PBS, incubated for 45 min with an anti-mouse FITC-conjugated secondary antibody, washed in PBS and mounted with Vectashield. Quantification of the intensity of 5′-FU fluorescence was determined within the nuclear perimeter in at least 100 GCs from three animals of each karyotype.

### 5.4. Conventional and Immunoelectron Microscopy

To analyze the ultrastructure of GCs, three TS and control mice were perfused with 3% glutaraldehyde in 0.1 M phosphate buffer pH 7.4. Their brains were removed and post-fixed overnight with the perfusion buffer. Then, coronal brain slices (300 μm thick) were obtained using a vibratome, and small fragments of the GCL of the hippocampus were dissected out. The processing of the hippocampal samples was performed following the procedure previously reported by [25].

For immunoelectron microscopy of coilin, three CO and TS mice were perfused with 3.7% paraformaldehyde in 0.1 M phosphate buffer. Small fragments of the GCL of the hippocampus were isolated from 300 μm-thick slices. The samples were processed following the protocol previously described by Pena et al. [37]. The samples were examined with a Phillips EM-208 electron microscope operated at 60 kV.

### 5.5. Ag-NOR Staining

Active chromosomal NORs were revealed in squashes of GCs by a silver impregnation method following the protocol described by [22]. Images were obtained using a Zeiss Axioskop 2 Plus microscope with a 100X objective. Ag-NOR spots were quantified in the nuclei of at least 100 GCs from three animals of each group.

### 5.6. Western Blotting

Mice were euthanized by decapitation and their hippocampi were dissected and stored at −80° C. Six hippocampi from TS and CO mice were lysed and homogenized as previously described [68]. The homogenates were boiled for 10 min and sonicated for 5 cycles of 30 s On/Off at 4° C using a Bioruptor Plus (Diadode) and left on ice for 20 min. The total protein content of each sample was determined following the protocol described by [88]. Identical amounts of protein from each sample were loaded on a 15% sodium dodecyl sulfate-polyacrylamide gel, electrophoresed and transferred to a polyvinylidene difluoride (PVDF) membrane. Blots were incubated with a polyclonal rabbit anti-acetyl histone H4 (06-598, Upstate), polyclonal rabbit anti-tri-methylated histone H4 at K20 (07-463, Upstate) and polyclonal rabbit anti-nucleolin (ab22758, Abcam) overnight at 4° C. To ensure equal loading, the blots were reproved using a mouse monoclonal anti-GAPDH antibody (6C5) (1:2000; Santa Cruz Biotechnology, Santa Cruz, CA, USA). After extensive washing, immunoblots were developed with goat anti-rabbit IRDye 680RD antibody or goat anti-mouse IRDye800CW (1:10,000; LI-COR Biotechnology, Lincoln, NE, USA)

Protein bands were detected using a LI-COR ODYSSEY IR Imaging system V3.0 (LI-COR Biotechnology) and quantified following the protocol described by [68].

### 5.7. Real Time Quantitative PCR (qRT-PCR)

For qRT-PCR analysis, six control and TS mice were anesthetized and decapitated. Their hippocampi were removed, rapidly frozen and stored at −80 °C. RNA was isolated using RNeasy Plus Mini Kit (Qiagen, Hilden, Germany). RNA was reverse-transcribed using RevertAid H Minus First Strand cDNA Synthesis Kit (Thermo Fisher, Vilnius, Lithuania) and cDNA concentration was measured in a spectrophotometer (Nanodrop Technologies ND-1000) and adjusted to 1 µg/µL. The rRNA expression of 45S precursor, 18S 5′junction intermediary and mature 18S and mRNA expression of genes encoding nucleolar components nucleolin *(Ncl)*, fibrillarin *(Fbn1)* and UBF *(Ubtf)* was determined by RT-qPCR using gene-specific SYBRGreen (TaKaRa)-based primers. Each individual sample was performed in triplicate. The threshold cycle (Ct) was determined and the results were normalized to the housekeeping gene GAPDH. Relative expression was calculated according to the 2-(ΔΔCt) equation [89]. SYBRGreen-based specific primers for murine RNAs were: for MeCP2 5′-gtgaaggagtcttccatacggtc-3′ and 5′-tctccttgcttttacgccc-3′, for Histone H1 5′-cgccgactcccagatcaagt-3′ and 5′-gaccttttgggctcatcgcc-3′, for 45S 5′-gaacggtggtgtgtcgtt-3′ and 5′-gcgtctcgtctcgtctcact-3′, for 18S 5′junction precursor 5′-cgcgcttccttacctggttg-3′ and 5′-ggagaggagcgagcgacc-3′, for mature 18S 5′-gatggtagtcgccgtgcc-3′ and 5′-ccaaggaaggcagcaggc-3′, for 28S 5′junction precursor 5′-cctcctcgctctcttcttcc-3′ and 5′-cctgttcactcgccgttact-3′, for mature 28S 5′-gtgacgcgcatgaatgga-3′and 5′-tgtggtttcgctggatagtaggt-3′, for nucleolin 5′-attggggagggaagggaagt-3′ and 5′-tcagcacttcgagttgaagca-3′, for fibrillarin 5′-tctgttccctggagagtctg-3′ and 5′-gggttccaggctctgtactc-3′, for UBF 5′-ccgcgcagcatacaaagaat-3′ and 5′- gtggtccggctagacttgg-3′, for nucleophosmin 5′-tcggctgtgaactaaaggct-3′ and 5′-gcccctgctcctaaactgac-3′, for coilin 5′-ccgaggtggtggaatacgct-3′and 5′-aggccagaggtcagatccaga-3′ and for GAPDH 5′-aggtcggtgtgaacggatttg-3′ and 5′-tgtagaccatgttagttgaggtca-3′.

### 5.8. Statistical Analysis

For comparisons, data was analyzed using Student’s *t-*test. All analyses were performed using SPSS (version 22.0, Chicago, IL, USA) for Windows.

## Figures and Tables

**Figure 1 ijms-22-01259-f001:**
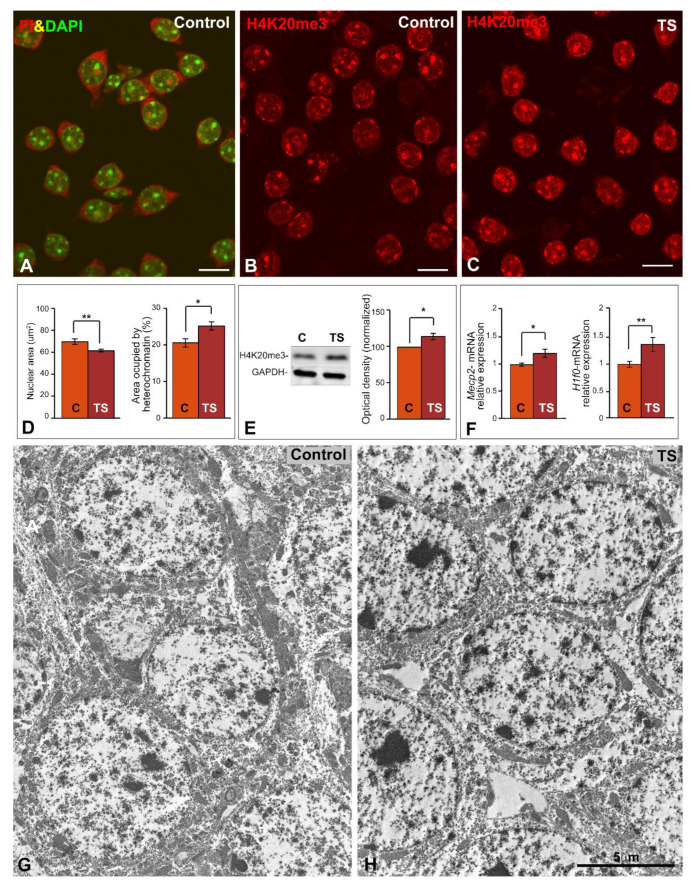
GCs of TS mice show reduced nuclear size and increased heterochromatinization. (**A**) Representative example of dissociated dentate gyrus GCs from a control mouse co-stained with DAPI (DNA staining) and propidium iodide (PI, RNA staining). Note the well-preserved neuronal perikarya, illustrating the concentration of RNA in the cytoplasm and nucleoli, and the prominent heterochromatin clumps counterstained with DAPI. (**B**,**C**) Histone H4K20me3 immunolabeling of dissociated GCs from control (**B**) and TS mice (**C**). Note the higher nuclear fluorescent signal of this trimethylated histone in TS GCs. Scale bars: 5 µm. (**D**) Morphometric analysis of the nuclear size and the nuclear surface area occupied by histone H4K20me3-positive clumps in control and TS GCs. Partial trisomy in TS mice associates with a reduction of nuclear size and a parallel increase of repressive heterochromatin domains. (**E**) Up-regulation of histone H4K20me3 expression in the hippocampi of TS mice relative to control animals as shown by Western blotting. GAPDH was used as loading control. (**F**) RT-qPCR determination of *Mecp2* and *H1f0* gene expression in hippocampal RNA extracts from control and TS mice. Both genes are up-regulated in the TS mice. (**D**–**F)**: Data is presented as means + SEM of three independent experiments. (**G**,**H**) Electron micrographs of GCs from control (**G**) and TS (**H**) mice. Note the prominent heterochromatin clumps in TS GCs. Scale bar: 5 µm. *: *p* < 0.05, **: *p* < 0.01. C = control mice; TS = Ts65Dn mice.

**Figure 2 ijms-22-01259-f002:**
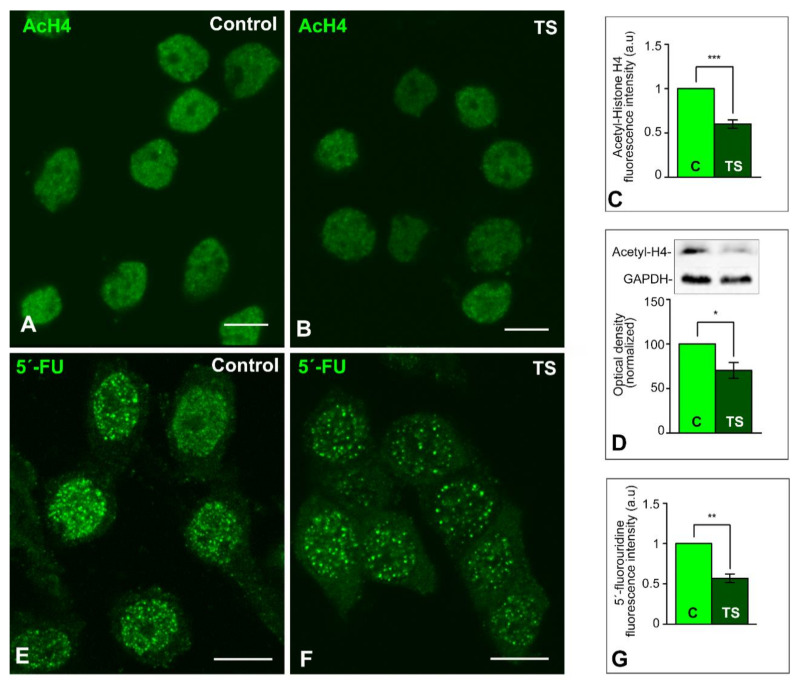
Global transcription rate is reduced in the GCs of TS mice**.** (**A**,**B**) Nuclear immunostaining of the acetylated histone H4 in dissociated GCs from control (**A**), and TS (**B**) mice. This acetylated histone appears diffusely distributed throughout the nucleus, excluding the nucleolus, with a higher intensity of fluorescent signal in control GCs. Scale bars: 5 µm. (**C**) Densitometric analysis reveals a significant increase of acetylated histone H4 signal intensity in control GC nuclei as compared with TS ones. (**D**) Down-regulation of acetylated histone H4 expression in the hippocampi of TS mice relative to control animals as shown by Western blotting. GADPH was used as loading control. (**E**,**F**) In situ transcription assay with 5′-fluorouridine illustrates the nuclear incorporation of this halogenated precursor into nascent RNA from control (**E**), and TS (**F**) GCs. Note the higher abundance of microfoci of high fluorescent signal intensity of RNA (transcription factories) in control GC nuclei. Scale bars: 5 µm. (**G**) Densitometric analysis reveals a significant increase of 5′-fluorouridine incorporation signal intensity in control GC nuclei as compared with TS ones. C, D, G: Data is presented as means + SEM of three independent experiments. *: *p* < 0.05, **: *p* < 0.01, ***: *p* < 0.001. AcH4 = acetylated histone H4; C = control mice; 5′-FU = 5′-fluorouridine; TS = Ts65Dn mice.

**Figure 3 ijms-22-01259-f003:**
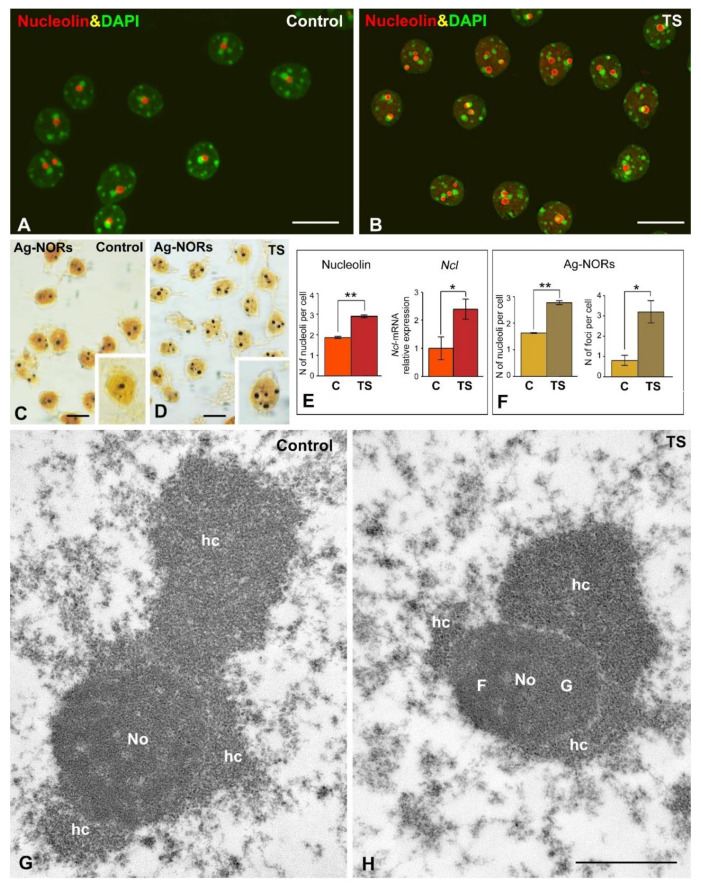
Trisomy causes an increase of nucleoli and Ag-NORs number in the GCs of TS mice. (**A**,**B**) Nucleolin immunolabeling counterstained with DAPI of dissociated GCs from control (**A**), and TS (**B**) mice. Note the apparent increase in the number of nucleoli in TS GCs. (**C**,**D**) Dissociated GCs from control (**C**), and TS (**D**) mice stained with the silver nitrate procedure for Ag-NOR proteins. Whereas large silver-impregnated spots correspond to Ag-NOR-containing nucleoli, the smaller foci (insets) belong to extranucleolar Ag-NORs. Scale bars: A-D: 5 µm. (**E**) The quantitative analysis of the mean number of nucleolin-positive nucleoli per GC shows an increase in TS GCs relative to control ones. This significant rise in the number of nucleoli correlates with a parallel increase in the expression levels of *Ncl* mRNA. (**F**) The quantitative analysis of the distribution of Ag-NORs in nucleoli and extranucleolar tiny foci reveals an increase in both the number of nucleoli and extranucleolar Ag-NOR foci in TS GCs. E and F: Data is presented as means + SEM of three independent experiments. (**G**,**H**) Electron micrographs of nucleoli (No) from control (**G**), and TS (**H**) GCs surrounded by prominent masses of perinucleolar heterochromatin (hc). Both nucleoli exhibit a predominance of the granular component (**G**). F: fibrillar component. Scale bar: 1 µm. *: *p* < 0.05, **: *p* < 0.01. Ag-NORs = argyrophilic nucleolar organizer regions; C = control mice; G = granular component; F = dense fibrillar component; hc = heterochromatin; No = nucleolus; TS = Ts65Dn mice.

**Figure 4 ijms-22-01259-f004:**
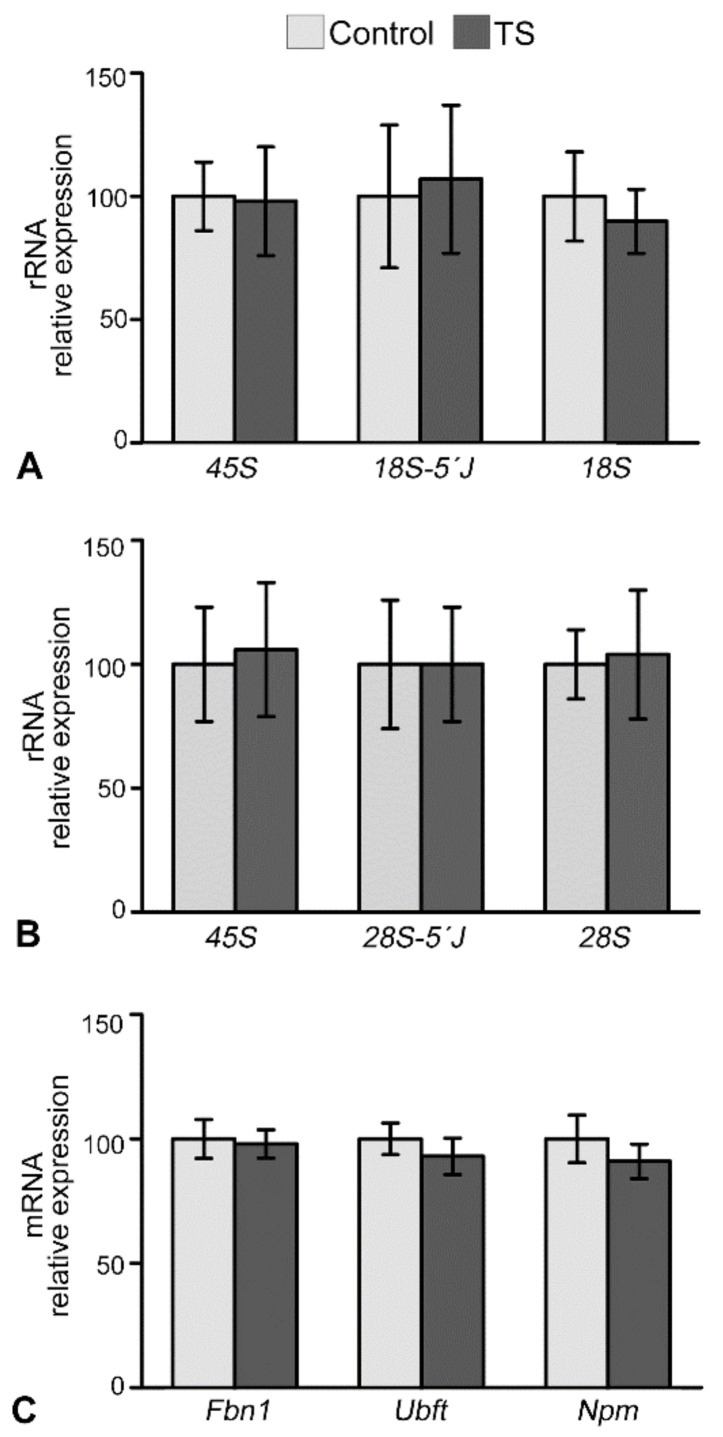
Ribosomal and nucleolar proteins genes expression is preserved in the GCs of TS mice**.** (**A**,**B**) RT-qPCR of the expression levels of pre-rRNA 45S, 18S 5′ and 28S 5′ junction precursors and mature 18S and 28S rRNAs in hippocampal RNA extracts from control and TS mice. Non-significant changes in all studied categories of rRNAs were detected in TS mice as compared to wild-type. (**C**) Expression levels of *Ubtf, Npm1,* and *Fbn1* mRNAs encoding three key nucleolar proteins, UBF, nucleophosmin, and fibrillarin. Non-significant differences between control and TS samples were observed in the three mRNAs analyzed. Data is presented as means + SEM of three independent experiments.

**Figure 5 ijms-22-01259-f005:**
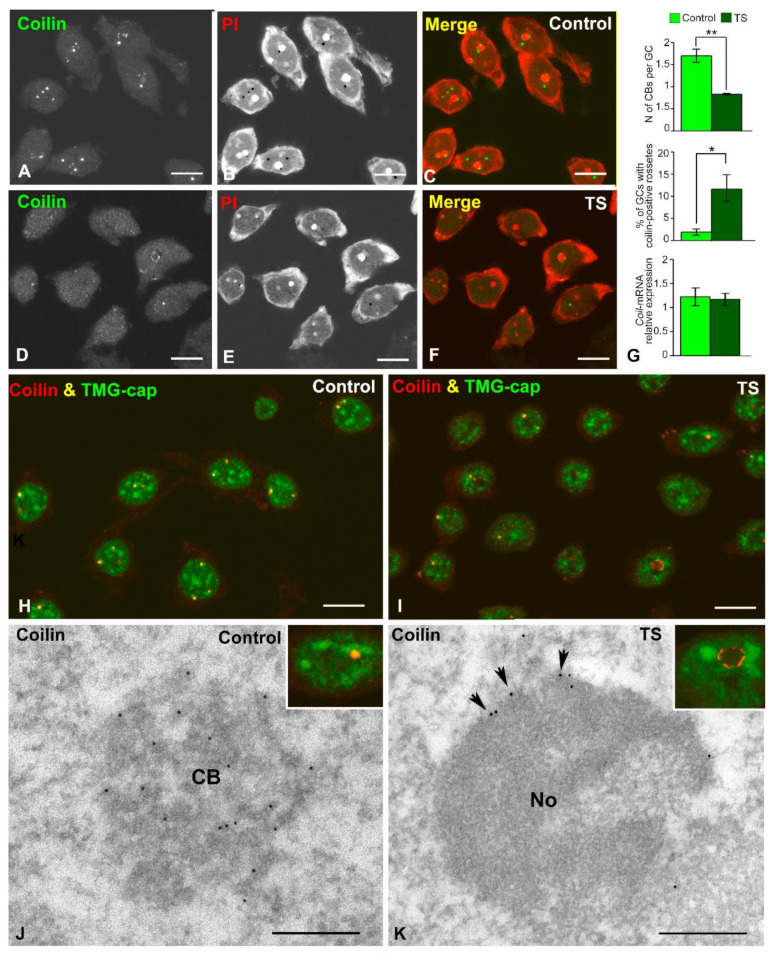
The GCs of TS mice present reduced CB number and coilin redistribution. (**A**–**F**) Co-staining for coilin, for demonstration of CBs, and PI, for the RNA-rich structures, in dissociated GCs from control (**A**–**C**), and TS (**D**–**F**) mice. Note the apparent reduction of CBs in TS GCs relative to control ones. Scale bars: 5 µm. (**G**) Quantitative analysis of the mean number of CBs per GC and the proportion of GCs containing coilin-positive perinuclear caps (“rossetes”). TS GCs exhibit a significant reduction of CBs and a significant parallel increase of perinuclear “rossetes”. This nuclear reorganization of coilin in TS GCs was not associated with significant changes in the expression level of the *Coil* gene. Data is presented as means + SEM of three independent experiments. (**H**,**I**) Double immunolabeling for coilin and TMG-cap (a marker of spliceosomal U snRNAs) reveals the typical colocalization of coilin and splicing factors in CBs and the absence of any detectable TMG-cap signal in the coilin-positive perinucleolar caps. Scale bars: 5µm. (**J**,**K**) Immunogold electron microscopy for the detection of coilin in typical CBs (**J**, scale bar: 250 nm) and a perinucleolar cap (arrow heads in panel **K**, scale bar: 500 nm). Insets illustrate the confocal microscopy counterpart of a double immunofluorescence for coilin and TMG-cap in GC nuclei. *: *p* < 0.05, **: *p* < 0.01. C = control mice; CB = Cajal body; No = nucleolus; PI = propidium iodide; TS = Ts65Dn mice.

## Data Availability

Not applicable.

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
