# Peer review of "Nuclear Reorganization in Hippocampal Granule Cell Neurons from a Mouse Model of Down Syndrome: Changes in Chromatin Configuration, Nucleoli and Cajal Bodies"

_ijms, 2021, doi:10.3390/ijms22031259_

Round 1

Reviewer 1 Report

The manuscript by Puente et al is a fascinating exploration of the architecture of cell nuclei in Down Syndrome, through a detailed morphological analysis of a mouse model Ts65Dn that harbors a triplication of 92 Hsa 21 genes through translocation. The findings from fluorescence light microscopy, electron microscopy and western blotting for histone tail markers provide clear evidence for increased heterochromatization and decreased overall transcription in hippocampal granule cell neurons. The data are nicely parametrized to allow statistical testing, which backs up these claims. Nucleoli are shown to increase in number and the nucleolar marker, nucleolin, increases. In contrast, the frequency of observation of Cajal bodies decreases. As nucleoli are associated with rRNA processing and Cajal bodies with pre-mRNA processing, the shift in prominence of these two bodies could reflect a shift in RNA metabolism in TS cells versus control and be relevant to disease. Overall, this is a rigorous and visually beautiful paper that should be published. I have a few suggestions for improvements to the paper below. A note to the authors: I apologize to the authors for my delay in submitting the review; I wish to explain that this was due to a month-long delay in getting a complete version of the manuscript from the journal.

  1. Interestingly, overall transcription appears to be decreased even though rDNA transcription is not significantly changed. In Figure 4, rRNA levels are measured between control and TS as well as mRNAs encoding 3 nucleolar and 1 Cajal body mRNA. None of these RNA levels are changed. This leaves us wondering which RNAs or RNA classes decrease in TS nuclei as measured by fluorouridine run-on. This is a difficult question to address without RNA-seq data, which may be envisioned as a follow-up study. However, this leaves some important open questions regarding the conclusions drawn. First, I would suggest measuring the levels of a few non-coding and coding RNAs that are not directly involved in these two bodies to enable the authors to comment on this aspect. I suggest RNA levels be normalized to one of the mitochondrial RNAs (transcribed in mitochondria), because these are likely to escape complications of nuclear architecture. Related to the previous point, I am wondering if the newly appearing heterochromatinized blobs are actually the chromosome harboring the translocation. If so, the transcription of the 92 triplicated genes may be down-regulated (normalizing their expression?) as well as of the other genes on the chromosome. If this were the case, it would reveal an important feature of the Ts65Dn model and raise new question about the effects of trisomy 21 on nuclear architecture.

  1. Note that current concepts of nuclear bodies include the expectation that they form on active transcription sites, with nucleoli forming on active rDNA and CBs forming on active U snRNA and histone gene loci. There are several papers on nucleoli (Karpen, Schaefer and Laird. 1988 Genes and Dev PMID 3149250; Frank and Roth MB. 1998 J Cell Biol. 140(6):1321-9; Falahati, H… and Wieschaus, E. (2016) Curr. Biol. 26, 277−285.) and more up to date publications on CBs that would be helpful to cite and discuss in this context (Arias-Escayola and Neugebauer 2018 Biochemistry; Machyna et al. 2014 Mol Cell; Wang Q, …. Dundr M. 2016. Nat Commun 7:10966.). The reduction of CBs in the Ts65Dn model could indicate reduced transcription of U snRNA and histone gene loci, which could be investigated or at least discussed.

  1. Minor: All of the figure legends would benefit from figure titles.

Reviewer 2 Report

Manuscript ID: ijms-1029383

Title: Nuclear reorganization in hippocampal granule cell neurons from a

mouse model of Down syndrome: Changes in chromatin configuration, nucleoli

and Cajal bodies

Authors: Alba Puente, María T. Berciano, Olga Tapia, Carmen Martínez-Cue,

Miguel Lafarga *, Noemí Rueda *

In this research, the authors studied the nuclear structure of hippocampal granule cell neurons from a mouse model of Down syndrome. They investigated the changes in chromatin configuration, nucleoli and Cajal bodies and proposed that these changes might correlate with the cognitive dysfunction characteristic of the TS mice. Overall, the results are interesting and might provide implications to the mechanistic studies of Down syndrome.

Suggestions to improve the quality of the paper:

The authors have observed changes in chromatin configuration and the number of nucleoli. Are there any relationship between these two events? That is, is the change of nucleolar number caused by the change in positions of rDNA due to chromatin reorganization or upregulated expression of nucleolin or other proteins involved in nucleolus assembly? Have the authors determined the protein level of nucleolin besides the Ncl mRNA level? Any explanations for the phenomena that Ncl expression was increased in Figure 2, while the global transcription was repressed in Figure 2?

Any explanations for the unchanged rRNA expression levels in the TS mouse model although the number of nucleoli were significantly changed? The results in Figure 4 were inconsistent with the statements in second paragraph of page 8.

Can the authors measure the status or activity of mRNA splicing in case that the morphology of Cajal bodies were altered in the TS mouse cells?

Considering that the nucleoli and Cajal bodies are also involved in the biogenesis of other small nuclear RNAs such as snoRNAs and scaRNAs, I suggest the authors to determine whether the biogenesis of these small non-coding nuclear RNAs as well as ribosome assembly was altered in the TS mouse model.

Is there any change in translation efficiency accompanied with the observed defects in transcription, splicing and number of nucleoli?

Figure legends, I suggest the authors to list the full names of the abbreviations used in the figures to facilitate the readers.
